

# Self-Learning Network-based segmentation for real-time brain M.R. images through HARIS

Parvathaneni Naga Srinivasu[1,*] and Valentina Emilia Balas[2,*]

[1] Department of Computer Science and Engineering, GITAM Institute of Technology, GITAM Deemed to be University, Visakhapatnam, Andhra Pradesh, India
[2] Department of Automation and Applied Informatics, Aurel Vlaicu University of Arad, Arad, Romania
[*] These authors contributed equally to this work.

## ABSTRACT

In recent years in medical imaging technology, the advancement for medical diagnosis, the initial assessment of the ailment, and the abnormality have become challenging for radiologists. Magnetic resonance imaging is one such predominant technology used extensively for the initial evaluation of ailments. The primary goal is to mechanizean approach that can accurately assess the damaged region of the human brain throughan automated segmentation process that requires minimal training and can learn by itself from the previous experimental outcomes. It is computationally more efficient than other supervised learning strategies such as CNN deep learning models. As a result, the process of investigation and statistical analysis of the abnormality would be made much more comfortable and convenient. The proposed approach's performance seems to be much better compared to its counterparts, with an accuracy of 77% with minimal training of the model. Furthermore, the performance of the proposed training model is evaluated through various performance evaluation metrics like sensitivity, specificity, the Jaccard Similarity Index, and the Matthews correlation coefficient, where the proposed model is productive with minimal training.

# INTRODUCTION

Medical imaging technology has been critical in medical diagnostics for accurately detecting the presence of malignant tissues in the human body. There are divergent imaging technologies that are available for diagnosis of abnormality among various organisms that includes X-ray technology, as stated by *Vaga & Bryant (2016)*, computed tomography (CT) technology, as stated by *Venkatesan, Sandhya & Jenefer (2017)*, magnetic resonance imaging (MRI) technology (*Chahal, Pandey & Goel, 2020*; *Srinivasu et al., 2020a*; *Srinivasu et al., 2020b*), and positron emission tomography (PET) scan (*Norbert et al., 2017*; *Schillaci & Urbano, 2019*). All the approaches mentioned above are non-invasive and capable of diagnosing malignant tissue efficiently. In many cases, imaging technology is aptly suitable for identifying abnormalities in the human body. It would help the physician provide better treatment and assist in planning the clinical procedure. The current study has focused on

Corresponding author
Valentina Emilia Balas,
balas@drbalas.ro

mechanizing a model capable of diagnosing the MR imaging and estimating the extent of the damage.

Medical imaging technology has upgraded stupendously to elaborate every minute and tiny ailment in the human body that could efficiently diagnosed disease at a significantly earlier stage. The proposed study primarily focused on tumor identification and volumetric estimation of the tumorous region in the human brain. There are numerous machine learning models available for accurately identifying the tumorous regions from MR images. Yet, most of these techniques are imprecise, process thirsty, and particularly semi-automated procedures used in tumor identification. To circumvent the limitations of the above methodologies, several automated techniques, including the conventional models like Genetic Algorithm (GA), as stated by *Srinivasu et al. (2020a)* and *Naga Srinivasu et al. (2020)*, Artificial Neural Networks (ANN), as said by *Wentao et al. (2020)* and Deep Learning techniques as mentioned by *Deepalakshmi et al. (2021)*, *Srinivasu et al. (2021)*.

As part of recognizing ailments, several stages, including pre-processing the MR images to address noise in the original MR images, such as Spackle noise, Poisson noise, and Gaussian noise, are introduced into the image at various stages during the process of rendering the image. And the pre-processed step will enhance the contrast and the texture of the image that assist in a faster and convenient way to recognize the malignant tissues. Once the noise in the image is pre-processed, the MR image is fed to remove the skull region. Then the MR image is segmented to recognize the malignant region, followed by the volumetric estimation for analyzing the impact of the damage. The results of the automated segmentation approaches are refined over multiple iterations for the precise outcome.

Generally, the medical images are segmented to locate the abnormal regions in the image based on texture-based information. The process of image segmentation plays a vital role in the identification of the malignant areas in the MR image; There are various semi-automated and automated ways of segmenting the MR images that could efficiently segment the images, but there is a considerable tradeoff between the accuracy and the computation efforts put forward by each of those approaches. Supervisory models yield better accuracy, but they need tremendous training data that needs more computational resources.

The K-Means algorithm is one such semi-automated approach that segments the MR Image based on the pre-determined number of segments as stated by *Alam et al. (2019)* and *Chahal, Pandey & Goel (2020)*. Thus, the proposed approach is one of the simple techniques for the segmentation of the images. But the major drawback of this approach is that the total number of segments is fixed well before the segmentation has begun. Therefore, if the *k* value is too large, it would lead to over-segmentation, and if the k value is too small, it will lead to under-segmentation of the image.

Segmentation of MR images is indeed possible using completely automated techniques such as thresholding, as stated by *Sowjanya, Rajesh & Satish (2018)* and *Kumar et al. (2015)*. The threshold-based method works comparatively better in the homogeneous image, but accuracy mainly depends on the approximated threshold value. A minimal deviation from the ideal value of the threshold results in an exponential variance in the segmented image.

The other predominantly used segmentation technique is through the region growing (*Dalvand, Fathi & Kamran, 2020*) strategy that can effectively handle the problem of over and under segmentation often encountered in K-Means-based approaches. The experimental studies on the region growing-based approach are proven to improve the sensitivity and specificity for precise identification of the malignant region in the human brain (*Punitha, Amuthan & Joseph, 2018*). But the seed point must be chosen appropriately, which needs tremendous efforts in recognizing the optimal seed point. If the initial seed points are inappropriately chosen, the entire process of segmentation of the image goes wrong, and also this approach needs computationally more effort.

*Sheela & Suganthi (2020)* and *Mathur, Mathur & Mathur (2016)* proposed Edge Based Segmentation for image segmentation, a relatively simplistic approach. But in the suggested edge-based approach, the image must be pre-processed rigorously to elaborate the edge-related information, which involves high computational efforts. *Pandav (2014)*, *Liang & Fu (2018)* and *Kornilov & Safonov (2018)* have proposed watershed-based image segmentation that effectively addresses limiting the number of segmented areas identified using edge information by marker-controlled watershed-based segmentation. The process of limiting the segments will assist in avoiding the over-fitting problem that is experienced in the majority of the user intervened models and the supervised models. But in watershed-based segmentation, considerable efforts are needed during pre-processing phase to separate the foreground and background regions in the image. *Sudharania, Sarma & Prasad (2016)* have proposed morphological based segmentation that exhibits an exceedingly high accuracy in less intensity MR images. But it needs several iterations to converge to a better solution, which needs more computational efforts.

*Jude, Anitha & Balas (2015)*, *Madallah, Muhammad & Muhammad (2020)* and *Verma, Agrawal & Sharan (2015)* have proposed fuzzy C means (FCM) based MR image segmentation as a highly accurate, well-founded, and rapid approach that effectively handles the uncertain situation of allocating pixels among multiple segments through distributing pixels to the appropriate segment depending on the membership value determined at each iteration. In fuzzy-based segmentation, the pixels are assigned to their corresponding region based on the membership function value that lies in a range of 0 to 1. The value 1 indicates the corresponding pixel is more likely to be associated with the corresponding segment. The biggest problem with the FCM technique is determining the membership value at each iteration for all pixels in the image that need additional processing efforts. In addition, the adjustment of the bottom and upper approximated for the randomness is complex though FCM.

*Al-Shamasneh et al. (2020)* have suggested an MR image segmentation through contour based segmentation that efficiently recognizes various brain tissues for both homogeneous and heterogeneous malignant cases. But it is not very efficient for noisy images and high-intensity images. *Li et al. (2018)* have proposed a level set centric technique for the MR image segmentation based on the pre-approximated threshold value. The level set method is a very complex approach, and the approximated threshold value determines the efficiency of the approach's segmentation.

*Wang, Li & Liu (2019)* and *Diaz & Boulanger (2015)* have proposed atlas-based segmentation for MR images as a straightforward approach that employs segmentation of the MR images. It is computationally faster than used labeling, and it is independent of the deformation model. The suggested approach merges the intensity template image and segmented reference image to register to further segment the image. In the atlas-based approach, choosing the initial seed point is crucial as the entire segmentation scenario is based on the seed point selection. However, the efficiency of this approach is based on the precision of the topological graph.

*Venmathi, Ganesh & Kumaratharan (2019)* and *Liu et al. (2018)* have proposed the Markov random field (MRF) to segment the MR image through the Gaussian mixture model that supports incorporating the neighboring pixels association in practical and mathematical perception. The method works on textured-based information. Markov random field-based approach for the segmentation of the image includes spatial information that aids in normalizing noise and overlapping the neighboring regions. *Javaria et al. (2019)* have suggested an approach that used the spatial vector alongside Gabor decomposition that distinguishes the malignant and non-malignant tissues in the MR image of the Bayesian Classifier. Despite the accuracy, MRF needs more computational efforts and the process of picking the parameters systematically.

*Varuna & Kumar (2018)*, *Gibran, Nababan & Sihombing (2020)* have suggested a probabilistic neural network (PNN) in combination with Learning Vector Quantization (LVQ) that assists in reducing the computational time by optimizing all the hidden layers in the proposed method. The region of interest that must be recognized for designing the network must be done carefully, as image segmentation quality depends on the exactness of the region of interest. *Sandhya, Babu & Satya (2020)*, *Mei et al. (2015)* and *De Guo (2015)* have suggested a self-organized map (SOM) based algorithm for segmentation of the MR image that includes the spatial data and the grey level intensity information in the segmentation of the image. It is outstanding in separating the malignant tissues. But the SOM engine has to be rigorously trained for better accuracy. The quality of the image segmentation is directly dependent on the training set. Mapping is one of the complicated tasks in a SOM-based approach.

*Havaei et al. (2016)*, *SivaSai et al. (2021)* and *Wentao et al. (2020)* have proposed a deep neural networks-based approach that is computationally efficient and highly precise in determining the abnormality from the medical image. Yet, the primary problem is that the implementation procedure and machine must be computationally efficient with adequate processing resources to perform the image segmentation in a reasonable time, which is not always technically feasible.

*Sachdeva et al. (2016)* have suggested multiple hybrid approaches for the segmentation of the MR images. The multiclass categorization of malignant tissues is done efficiently, and high accuracy is attained through machine learning and soft computing techniques. Pulse coded neural network (PCNN) is a technique used in coherence with the semi-automated methods for better segmentation. While segmenting the MR image, the region of interest could be perceived as a region growing approach that selects the initial points assumed as the seed points in the earlier stages. Secondly, a feed forward back neural network (FFBNN)

selects the seed points that send back the input until the input turns uniform. *Qayyum et al. (2017)* have attained multiple sub-images with multi-resolution data by employing a stationary wavelet transform (SWT). The spatial kernel is being applied over the resultant sub-images to locate the demographic features. With the help of extracted features and the stationary wavelet transform coefficient, the multi-dimensional features are built. The identified features and coefficients as the input to the self-organized map, linear vector quantization, are finally used to refine the results.

*Srinivasu, Rao & Balas (2020b)* has proposed a twin centric genetic algorithm with a social group optimization that has produced a precise outcome in tumor identification from the brain MR images. The twin-centric GA model is comparatively faster than the conventional GA approach, with a faster crossover rate that results in a new segment. The mutation operation is performed based on the fitness value to reform them with other strongly correlated regions in the image. The outcome of the Twin GA is refined through the social group optimization approach, which has refined the outcome through selecting the appropriate features in the image for the segmentation. *Dey et al. (2018)* has experimented with the SGO approach in fine-tuning the outcome of the classification. The current approach is comparatively faster, but in performing the two-point crossover, there is a chance of diverging from the optimal number of regions and may end up with the over-fitting issue. The execution of two high computational algorithms needs a significant computational effort to segmentation the MR image.

The main objective is to formulate a mechanism that can efficiently segment the real-time image into multiple regions based on the available features by minimizing the computation efforts. The proposed approach is a self-trained strategy that upskills the algorithm with some pre-existing real-time scenarios. Previous experimental results show that the proposed SLNS algorithm can differentiate the skull region from the brain tissues in the MR image without any external pre-processing algorithm. It can quickly extricate brain tissues from the non-brain tissues through the available feature set. The proposed model is robust in handling the images with an acceptable noise level, and it needs less computational effort for training the model. The experimentation has been performed to evaluate the accuracy of the proposed approach, and the upshot seems to be promising.

## SELF-LEARNING NETWORK BASED REAL-TIME SEGMENTATION

Cognitive technology in real-time image segmentation is a multidisciplinary technique that is an intrinsic aspect of fully convolutional neural networks. Fully convolutional neural networks are widely utilized in real-world settings to successfully handle 2D images. The concept of semantic segmentation has been used in coherence with multi-objective function-based algorithms for better results. The weak learning network is based on the partial training of the algorithm and the tuning of the algorithm so that it would be able to differentiate among multiple classes. In addition, the algorithm is capable of training itself for better efficient segmentation of the image. In every stage of execution for evaluating the tumor's region from the MR image, the proposed method would segment the image based

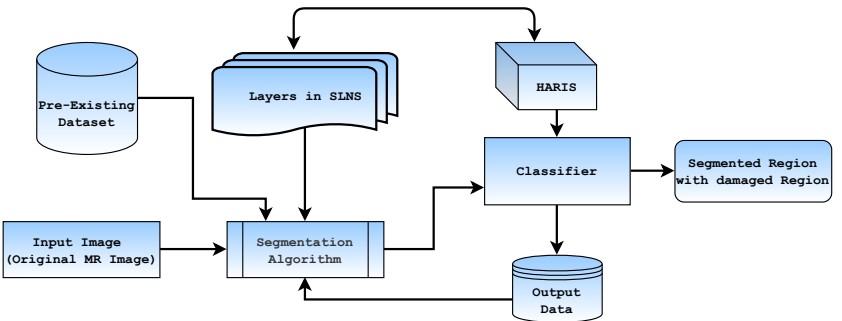

**Figure 1** **Represents the block diagram of SLNS based approach.**

on the trained information. The resultant segmented imaged would be further refined in further network layer's multi-objective functions.

## PARTIAL TRAINING OF THE ALGORITHM

In contrast to its competitors, the suggested methodology is a partly trained strategy that does not need extensive training. As shown by *Xiao et al. (2015)*, the recommended strategy involves training the original image with an acceptable quantity of noisy data to ensure that the algorithm can address noise images. Additionally, the noise picture is specified in the labels. The binary identifier notion is used to distinguish a noisy picture from a non-noise labeled image (*Misra et al., 2016*). It has proposed the idea of independently labeling the noise class. The role of differentiating the image-based noise classes is significantly essential. The image could be easily determined among the noise levels to process the images based on the noise variance.

The model is trained to distinguish brain tissues from non-brain tissues such as brain fluids, the skull region, the thalamus, and the brainstem, which must be isolated from brain tissues for improved segmentation and evaluation of the tumor's extent. However, whenever a new image is fed into the algorithm as input, post segmentation, the image is fed into the network's self-learning layer as input for further images.

## SELF-LEARNING NETWORK-BASED SEGMENTATION

In the automated segmentation of the brain MR image, the crucial thing is to set up training sets that are large enough to train and validate the model. As a result, it is challenging to obtain a self-sufficient dataset for both trains and validating with ground facts that are associated with the imaging data concerning to abnormality for cross validation. On the other hand, self-learning models need a relatively smaller training and validation set. Furthermore, the experimental results used in self-training the model ensure better results in future experimentation based on its previous experimental knowledge. The complete flow diagram of the proposed framework can be seen in Fig. 1 stated below.

## LAYERED ARCHITECTURE OF SLNS

The layered architecture of the proposed self-learning network-based segmentation (SLNS) approach includes the convolutional layers that would perform various challenging tasks like pre-processing of the image to remove the image's noise by considering the adaptive bilateral filter for pixels that surround the actual corresponding pixel. The image is then further processed to remove the skull region in the next successive layer in correlation with the other connected layers. Next, the image is segmented based on the pixel's intensity value, representing the region's texture. Finally, the classifier is used to dissimilate the non-brain tissues and damaged region in the human brain.

The suggested methodology trains the segmentation algorithm using a self-learning module that gets adequate information from the preset segmentation dataset and previous experimental results. Additionally, each convolutional layer consists of a variety of kernels, including an adaptive bilateral filter with sublayers that determine the appropriate number of clusters and cluster centroids, Sobel filter that is used in the edge detection, and other 2D convolution filters that are generally $(3 \times 3)$ in size which slides across the original input image. Over numerous epochs, the model is trained over multiple iterations to identify all the key features in the input image. The image is refined at the earlier stages using an adaptive bilateral filter that processes the image that normalizes the noise like Gaussian noise and Poisson noise during image acquisition. The method of noise reduction is carried forward through the following equations

$$AB_F = \alpha \times \sum_p p_b\left(\left|\left|x-y\right|\right|\right) \times p_l\left(\left|\left|i_x - i_y\right|\right|\right) \tag{1}$$

From the above equation, the $\alpha$ value is determined by the contra harmonic mean of the neighboring pixels determined in Eq. (2) stated below, Pb represents the pixel belongingness, and $p_l$ represents the pixel likeliness.

$$\alpha = \frac{i_1^2 + i_2^2 + i_3^2 + \ldots + i_n^2}{i_1 + i_2 + i_3 + \ldots + i_n} \tag{2}$$

The values of the pixel belongingness (Pb) have been demonstrated in Eq. (3) mentioned below

$$P_b = \frac{\sum_p c_p(p)^x p}{\sum_p c_p(p)^x} \tag{3}$$

where cp determines the membership coefficient, and p designates the pixel, and x designates the fuzzier metric of belongings. The value of the pixel likeliness(pl) is designated through Eq. (4) stated below

$$p_l = p(s) \times p\left(\frac{cp}{s}\right) / \sum p(k) \times p\left(\frac{cp}{k}\right) \tag{4}$$

From the above equation, the p(s) denotes the probability of likeliness with the segment s, and $p\left(\frac{cp}{s}\right)$ represents the conditional probability concerning segment s and $\sum p(k) \times p\left(\frac{cp}{k}\right)$ is generally expected for all the classes.

Heuristic Approach for Real-time Image Segmentation (HARIS) algorithm is used to segment the high dimensional images like the medical MR images to identify the abnormality from the MR images. HARIS approach incorporates two phases in the image segmentation process. In the initial phase, the optimal number of regions in the image are being assessed, which assists in accurately recognizing the regions in the image based on texture, intensity, and boundary region-related pixels. The techniques like the intraclass correlation and interclass variance are being considered for evaluation. The second phase of the HARIS approach is to identify the local best feature within the region following the presumed feature element that is presumed to be the global best. The selected feature is used in assembling the pixels as a region based on the feature identified.

The image is segmented based on the intensity level in the later stages by approximating the minimum number of segments to 23 from the previous experimental studies. The number of segments fitness has been evaluated through the formula stated below

$$\text{Obj}_{\text{fun}} = \left(\text{x} \times \frac{\text{Tot}_{\text{pix}}}{\text{pix}_{\text{seg}}}\right) + \left(\text{y} \times \frac{\text{T}_{\text{s}}}{\text{N}_{\text{r}}}\right) \tag{5}$$

In the above Eq. (5), x, and y are the deciding factors that control the proposed algorithm's accuracy and efficiency. x determines the inter-class variance, and y determines the intraclass variance stated through Eqs. (7) and (9).

The maximum interclass variance is determined by the Eq. (6) stated below

$$\sigma^2 \text{inter}_{\text{c}}(C_t) = \sigma^2 \text{total}(C_t) - \sigma^2 \text{prev\_inter}(C_t) \tag{6}$$

$$\sigma^2 \text{inter}_{\text{c}}(C_t) = \sum_{x=0}^{C_t-1} p(x) \sum_{x=C_t}^{Z-1} p(x) [\mu_1(C_t) - \mu_2(C_t)]^2 \tag{7}$$

Equation (7) is the elaborated version of the Eq. (6) and Ct in the above equation threshold value of the class that is evaluated through the fuzzy entropy-based thresholding (FET) approach. The variables $\mu_1, \mu_2$ That determines the means of the intensities of the image segment.

The fuzzy entropy-based (*Oliva, Elaziz & Hinojosa, 2019*) thresholding technique that evaluates how strongly a pixel is strongly correlated to a particular segment, the value of the threshold is being determined by the Eq. (8).

$$\text{FET} = \sum_{\text{ints}=1}^{\text{max}} \mu_{\text{tc}}(\text{ints}) \log 2 \mu_{\text{tc}}(\text{ints}) - \sum_{\text{ints}=\text{max}+1}^{255} \mu_{\text{ntc}}(\text{ints}) \log 2 \mu_{\text{ntc}}(\text{ints}) \tag{8}$$

From the above Eq. (8), the variables $\mu_{\text{tc}}, \mu_{\text{etc}}$ Are the variables representing the fuzzy membership concerning the image segments associated with the tumors class and non-tumors class. However, the equation is formulated concerning Shannon's entropy formulation. The value of the threshold(FET) would be greater than 0 and would lie below 255.

The value of the intraclass correlation is determined by Eq. (9),

$$I_{corl} = \frac{\sigma_s^2}{\sigma_s^2 + \sigma_i^2} \tag{9}$$

From Eq. (9), the variable $\sigma_s$ represents the standard deviation in concern to the given image segment, and the variable $\sigma_i$ represents the standard deviation concerning the entire image that is being considered.

## Adaptive structural similarity index

The adaptive structural similarity index (ASSI) metric is used to classify the pixel among the tumorous and non-tumorous regions. The structural similarity index relay on the likeness among the pixels among tumor regions. The structural similarity index relies on three factors: the structural parameter, luminance parameter, and contrast parameter. The index is determined as the product of the three factors mentioned above. This paper has proposed an adaptive structural similarity index that would assess the membership alongside the similarity index to make the outcome more realistic. The equation of the similarity index is determined through the equation stated below.

$$ASSI(p,q) = \omega \times x(p,q)\alpha \cdot y(p,q)\beta \cdot z(p,q)\gamma \tag{10}$$

$$x(p,q) = \frac{2\mu_p\mu_q + C_1}{\mu_p^2 + \mu_q^2 + C_1} \tag{11}$$

$$y(p,q) = \frac{2\sigma_p\sigma_q + C_2}{\sigma_p^2 + \sigma_q^2 + C_2} \tag{12}$$

$$z(p,q) = \frac{\sigma_{pq} + C_3}{\sigma_p\sigma_q + C_3} \tag{13}$$

$$\omega = \frac{\sigma_p^2}{\left(\sigma_p^2 + \frac{\sigma_e^2}{2}\right)} \tag{14}$$

The Eq. (10) mentioned above is used to assess the likelihood of the pixel that could be a part of the tumor region. The variable $\omega$ determines the probability that is multiplied by the product of the structural parameter presented in Eq. (11), luminance parameter presented in Eq. (12), and contrast parameter given in Eq. (13). The outcome of the proposed model is promising when compared to that of its counterparts. The ASSI is used alongside the trained models in the proposed self-learning model.

---

**Algorithm 1** Algorithm

---

**Date:** x(p,q)←Structural Parameter, y(p,q)←Luminance Parameter, z(p,q)← Contrast Parameter

**Input:** pixel(p,q) where p represents the row and the q represents the column

**Output:** Assess the correlation index

**pix(p,q)**← initialize the starting pixel
**while**(i<maximum_number_iterations)
    **for** each pixel in the input image, **do**
    Update x(p,q), y(p,q), z(p,q) and Ω
      Approximate the **Correlation Index**
        Select the regions in the image
        Determine the suitable category to assign
          **If** CI > Threshold, **then**
            Assign the corresponding pixel to the tumorous Region
          **else**
            Assign the corresponding pixel to the non-tumorous Region
         **end if**
    **end for**
**end While**

---

Figure 2 represents the proposed model's layered architecture. The convolutional layer, max-pool layer, flattening layer, and fully connected layers almost resemble the convolutional neural network model (*Farhana, Abu & Paramartha, 2020*). The convolutional layer in the proposed architecture is responsible for applying kernels' sequence to the input MR images for extracting the features from the image that assist in classifying various regions in the input image. The convolution layer identifies the pixels' spatial association based on the selected features informing the region, making abnormality identification easy. The pooling layer is generally next to the convolution layer, which is meant for reducing the special size of the outcome in the previous layer, which would resultantly minimize the number of parameters and features that are deliberated for further processing and to minimize the computational time. In the proposed model, the MAX pooling approach is considered in reducing the spatial size. Convolution and max pooling layers are used concurrently for numerous rounds until all regions are well-tuned, and features are recognized.

The flattening layer is the successive layer after the convolution and pooling layers. It is responsible for transferring the data into the 1-dimensional array used in the subsequent layered architecture. The fully connected layer in the proposed architecture uses the intensity features of the regions that are recognized as abnormal and normal in the considered MR images. Gated recurrent unit (GRU) is used in the proposed model to maintain the interpreted data of the previous experimental prediction. The same apprehension is used in the future in abnormality recognition from the MR image. A Heuristic Approach for Real-time Image Segmentation (HARIS) algorithm is used to

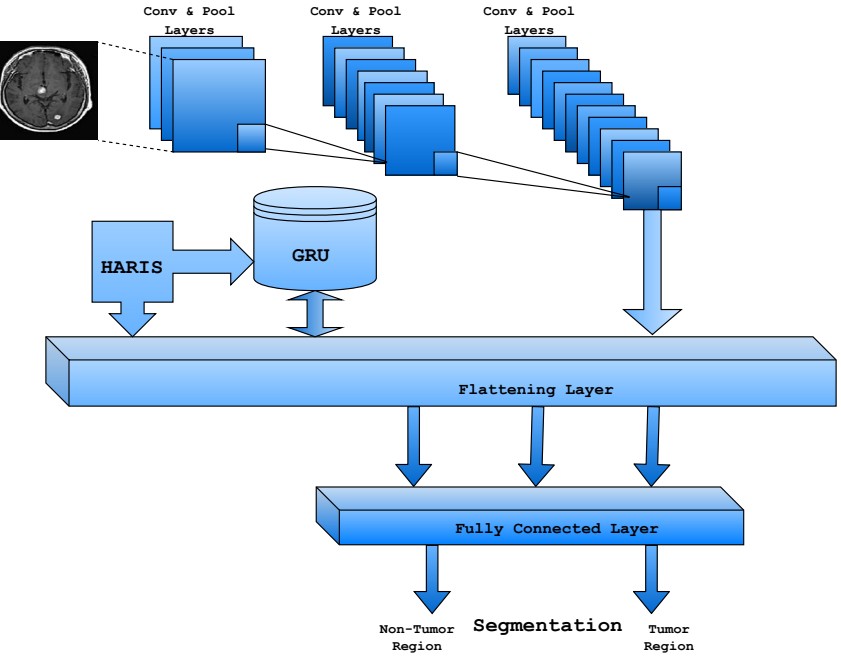

**Figure 2** Architecture diagram of the proposed SLNS model.

differentiate the tumors and non-tumors Region by identifying the appropriate pixels as the features for the prediction.

The gated recurrent unit is used to maintain the acquired knowledge from the previous outcomes. GRU is efficient, with comparatively fewer parameters needed to maintain the training data, (*Guizhu et al., 2018*; *Le, Kim & Kim, 2016*). GRU can address challenging tasks like vanishing gradient problems through the two gates, namely the reset gate and the update gate. HARIS works with both the fully connected and GRU layers to recognize the suitable pixels for the prediction. The cost function of the proposed model in concern to the tensors for the input image I(i,j) can be evaluated through the Eq. (15)

$$j\left(w, \beta; i, j\right) = \frac{\left\| f_{w,\beta}\left(i\right) - j \right\|^2}{2} \tag{15}$$

The variable $w$ represents the weight associated with layer $l$ and the variable $\beta$ represents the bias, $f_{w,\beta}\left(i\right)$ represents the kernel that is used in operating the elements. The error at the layer $l$ in the proposed model is determined through the Eq. (16)

$$e^l = \left(\left(w^l\right)^t e^{l+1}\right) \cdot f\left(x^l\right) \tag{16}$$

The $w^l$ is the weight that is associated with layer $l$ and the variable $e^{l+1}$ represents the error associated with layer $l+1$. The variable $f\left(x^l\right)$ is the activation function that is associated with the layer $l$ that is being determined by the rectified linear unit (ReLu) in the proposed model.

$$ReLu\left(x^l\right) = max(0, x) \tag{17}$$

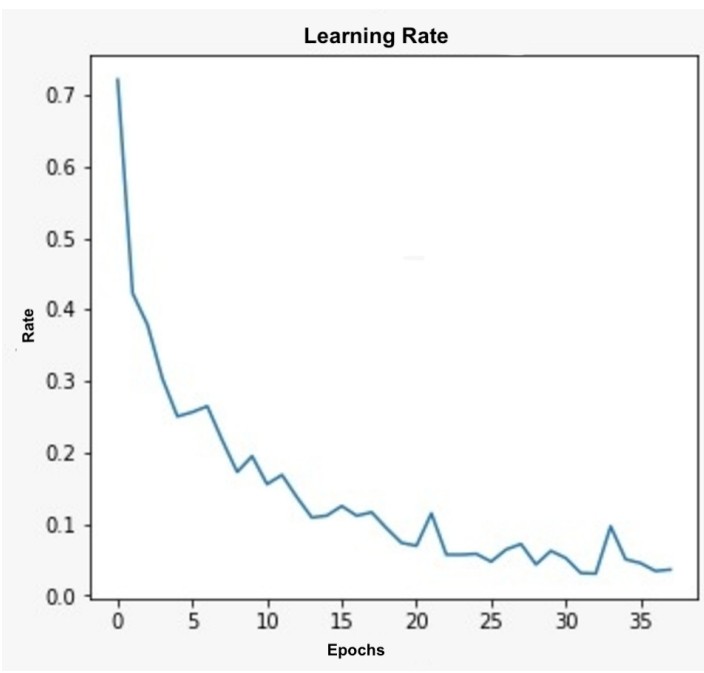

**Figure 3** The graph representing the Learning Rate.

The ReLu based activation function is linear for the greater than zero values, and it will be zero for all the negative values.

The learning rate of the proposed model is significant in assessing the performance of the model, which generally controls the weights in the network in concern to the loss gradience. The learning rate is presumed to be at an optimal level so that the model will move towards the solution by considering all the significant features in the prediction. The lesser learning rate presents the slower learning ability resulting in a delayed solution. On the other hand, the higher learning rate results in a faster solution that may ignore few features in the learning process. Figure 3 represents the learning rate of the proposed model across various epochs. Initially, with fewer epochs, the learning rate of the model is high; which implies the model is learning faster from the training data. As increase the epochs, the learning rate is saturated, which implies the model is learning only the new insights from the training data, the same scenario can be observed from Fig. 3. The saturation point for the learning rate in the proposed model is achieved at the epoch =43 and the iteration =187.

In the proposed self-learning centric segmentation model, the model will acquire knowledge from the earlier experimental outcomes. The learning rate is also dependent on the number of epochs that the model is designed to execute before it is evaluated. As the number of epochs increases, the learning rate of the model will move towards the saturation point, and we can observe the change in the learning rate untill that point. But increasing the number of epochs will result in consuming more computational efforts in the training process.

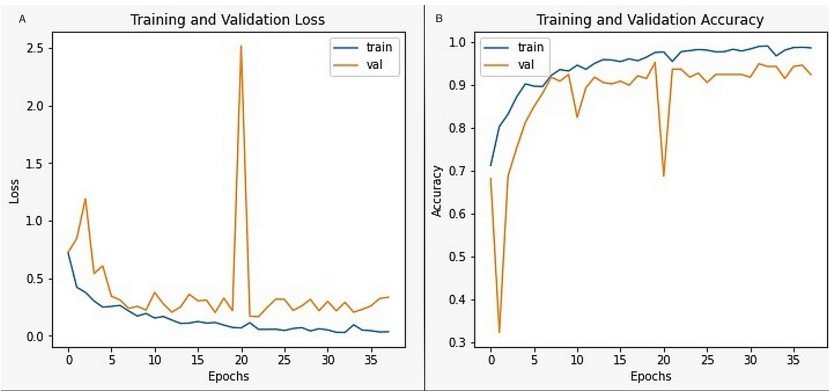

**Figure 4** **The image representing the hyper-parameters.** (A) Training and validation loss; (B) training and validation accuracy.

The other hyperparameters associated with the evaluation process include the loss and accuracy functions related to the proposed model's training and validation phase. Figure 4A represents the training, and the validation loss of the model, wherein graph after a certain number of epochs, approximately from 15, the training and the validation loss are close to each other, which determines that the proposed model is reasonably good in identifying the abnormalities from the MR images. When the validation loss is much higher than the training loss, it is assumed as the overfitting, which results in incorrect classification of the abnormal region. Underfitting is a situation in which the training loss is more than the validation loss that would result in poor accuracy of the model due to inappropriate selection of the features in the data. Thus, training and validation accuracy are the other parameters that are considered for evaluating the proposed model. Figure 4B represents the accuracy measures of the proposed model.

## EXPERIMENTAL RESULTS AND OBSERVATIONS

The experimentation has been carried forward over the real-time MR images acquired from the open-source repository LGG dataset acquired from The Cancer Genome Atlas (TCGA) captured from patients with acute glioma. The performance of the proposed approach has been evaluated through various metrics like Sensitivity, Specificity, Accuracy, Jaccard Similarity Index, and Matthews Correlation Coefficient that is assessed from the True Positive value that designates how many times does the proposed approach correctly recognize the damaged Region in the human brain as damaged Region and the True Negative that designates the how many times the proposed approach identifies non-damaged Region correctly and False Positive designates the number of times does the proposed approach identifies damaged Region in the brain as non-damaged Region and False Negative designate the count of how many times does the proposed approach mistakenly chosen non- tumors region as tumors region. Figure 5 presents the output screens of the proposed model.

The performance of the proposed self-learning network-based segmentation (SLNS) approach is evaluated against the other conventional approaches like twin-centric GA with SGO, HARIS approach, and convolutional neural network in concern to the performance evaluation metrics like Sensitivity, Specificity, Accuracy, Jaccard Similarity Index (JSI), Matthews correlation coefficient (MCC). Table 1 represents the experimental evaluations of the proposed model. The experimentation is conducted by executing the code repeatedly 35 times and scaled up for evaluating the confusion matrix. The model's accuracy is assessed with a standard deviation of ±0.015 in assessing the segmented image as a present by *Agrawal, Sharma & Bikesh (2019)*.

From Table 2, it can be observed that the proposed SLNS approach is outperforming when compared to its counterparts. In many of the cases, the proposed approach seems much better than CNN. The semi-trained approach needs comparatively fewer efforts than CNN at the same performance. In the computational efforts perspective, the proposed algorithms need almost the same execution time as the CNN algorithm but more time than the HARIS Algorithm-based segmentation approach. Fuzzy entropy-based MR image segmentation as experimented by *Rajinikanth & Satapathy (2018)*, *Chao et al. (2016)*. Figure 6 presents the comparative analysis of the various approaches.

The performance of the proposed model is assessed through the HARIS algorithm that decides the best possible number of segments to assist in identifying the abnormalities in the image, and it assigns the pixels to the segment by identifying the ideal pixel in the segment. The second objective function of the HARIS algorithm is replaced with the fuzzy membership assessment model, and the performance of the proposed model is being evaluated and presented in Table 3. The fuzzy membership is evaluated through Eq. (15) stated by *Vaidya, Metkewar & Naik (2018)*. The membership is evaluated as follows

$$membership = vertex_{pix} - \frac{vertex_{pix} - I'_{pix}}{1 - mem_{pix}(I'_{pix})} \qquad (18)$$

From Eq. (15), the variable $vertex_{pix}$ represents the vertices that are being considered in the image for processing. The variable $I'_{pix}$ represents the instance data that are the pixels, and the variable $mem_{pix}(I'_{pix})$ presents the minimum correlation value in concern to the MR image segment. The segment vertex is being assessed through the Eq. (16) stated below

$$vertex = vertex_{pix} + \frac{J'_{pix} - vertex_{pix}}{1 - mem_{pix}(J'_{pix})} \qquad (19)$$

From Eq. (16), $vertex_{pix}$ represents the vertex pixel in the region of the MR image. The variable $J'_{pix}$ represents the instance pixel in the segment and $mem_{pix}(J'_{pix})$ represents the membership value. It can be observed from Table 3. Thus, the fuzzy membership-based HARIS algorithm has slightly performed well than the traditional HARIS algorithm.

Figure 7 presents the assessed computational time consumed by various algorithms. It could be observed from the graphical values that the proposed approach SLNS seems to be much efficient. It is observed that SLNS consumes more time than the HARIS-based

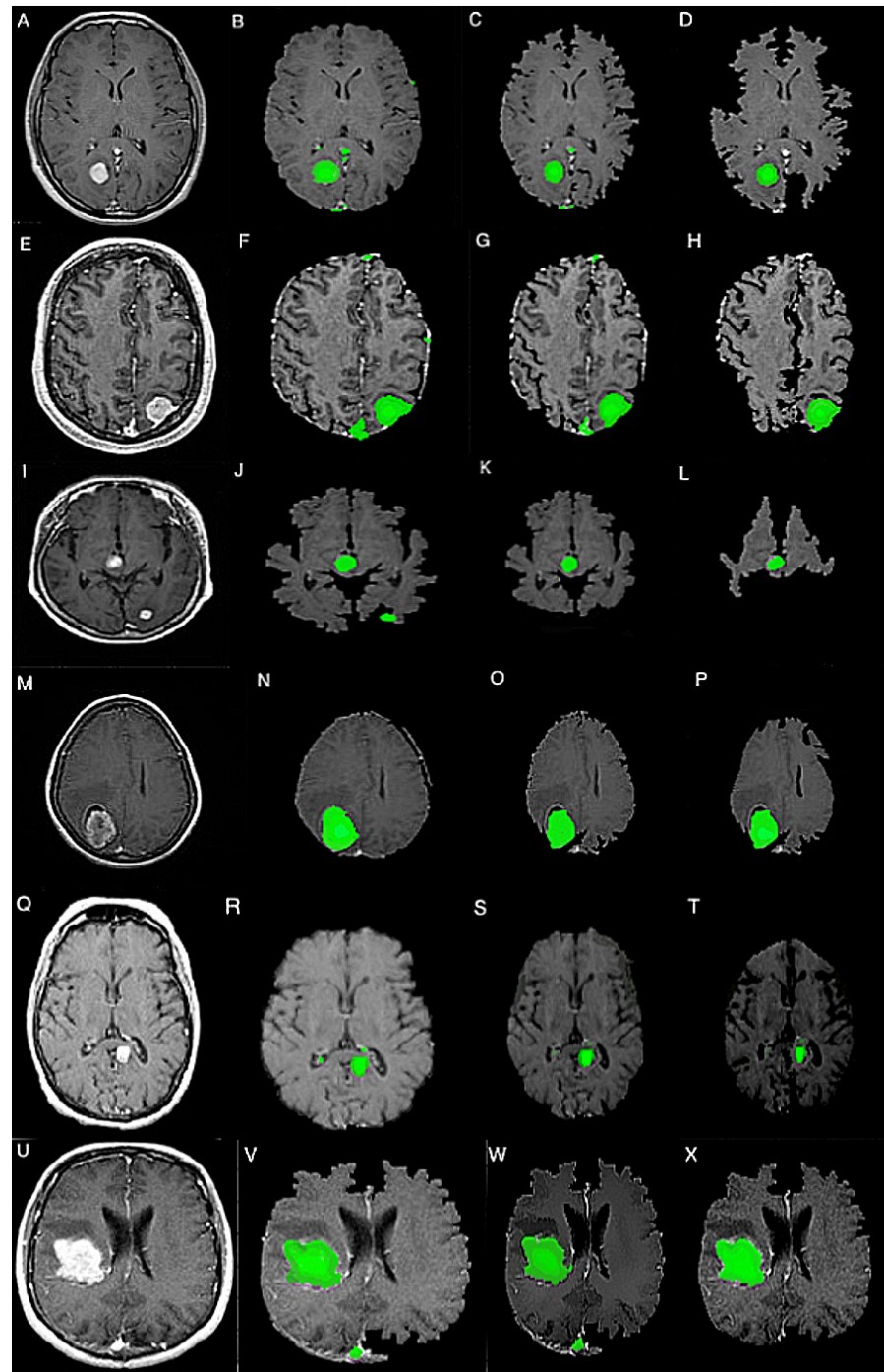

**Figure 5** Output screens of proposed SLNS approach (A–X).

**Table 1  The confusion matrix of the proposed model.**

|  | SLNS through HARIS | SLNS through Fuzzy Membership |
|---|---|---|
| True positive(%) | 82.85 | 77.14 |
| True negative(%) | 85.71 | 80.0 |
| False positive(%) | 17.15 | 22.86 |
| False negative(%) | 14.29 | 20.0 |

**Table 2  Performance analysis of the proposed approach.**

|  | Sensitivity | Specificity | Accuracy | JSI | MCC |
|---|---|---|---|---|---|
| Twin centric GA with SGO | 0.78432 | 0.83126 | 0.74232 | 0.83593 | 0.77523 |
| HARIS algorithm | 0.78973 | 0.84214 | 0.75147 | 0.84092 | 0.78614 |
| CNN approach | 0.79512 | 0.85424 | 0.76001 | 0.84912 | 0.79214 |
| Fuzzy entropy based segmentation | 0.99980 | 0.73060 | 0.85350 | 0.72770 | 0.84030 |
| SLNS Approach | 0.82829 | 0.87901 | 0.79932 | 0.86003 | 0.79714 |

**Table 3  The performance analysis with fuzzy component.**

|  | Sensitivity | Specificity | Accuracy | JSI | MCC |
|---|---|---|---|---|---|
| SLNS with Fuzzy membership | 0.81218 | 0.86294 | 0.779383 | 0.8612 | 0.82490 |
| SLNS with HARIS | 0.80241 | 0.85815 | 0.76212 | 0.84899 | 0.79001 |

approach, but from a precision point of view, the proposed SLNS has a trade-off. The computational time of CNN and SLNS is almost the same, and the computational lag in the proposed approach is because of self-training that needs additional efforts. The proposed model does not require any significant training, unlike the neural network-based models.

The proposed model's performance is assessed against the various existing algorithms like thresholding, seeded region growing, fuzzy C-means, and the artificial neural network models concerning the evaluation metrics like sensitivity, specificity, and accuracy presented by *Alam et al. (2019)*. The comparative analysis of the approaches with obtained values is shown in Table 4. The performances of various algorithms connected to the proposed SLNS have been assessed concerning the size of the tumor. It is observed that from the clinical evidence, the resultant outcomes have been proven to be better compared to their counterparts and the consequent of the proposed methods seems to very pleasing and very accurate. The tabulated values in Table 4 represent the resultant experimental values.

Table 4 represents the progress of the tumor growth, that is identified based on the texture information of the abnormal region as stated by (Naga et al., 2020). The abnormal region is classified as the tumour core (TC) that depicts the actual region of the tumor, and the enhanced tumour (ET) that depicts the recent enhancement that has taken place in the region of the tumor that is presumed to be the progress in the tumor. The whole tumour (WT) is the region that includes both the tumor core and the enhanced tumor regions.

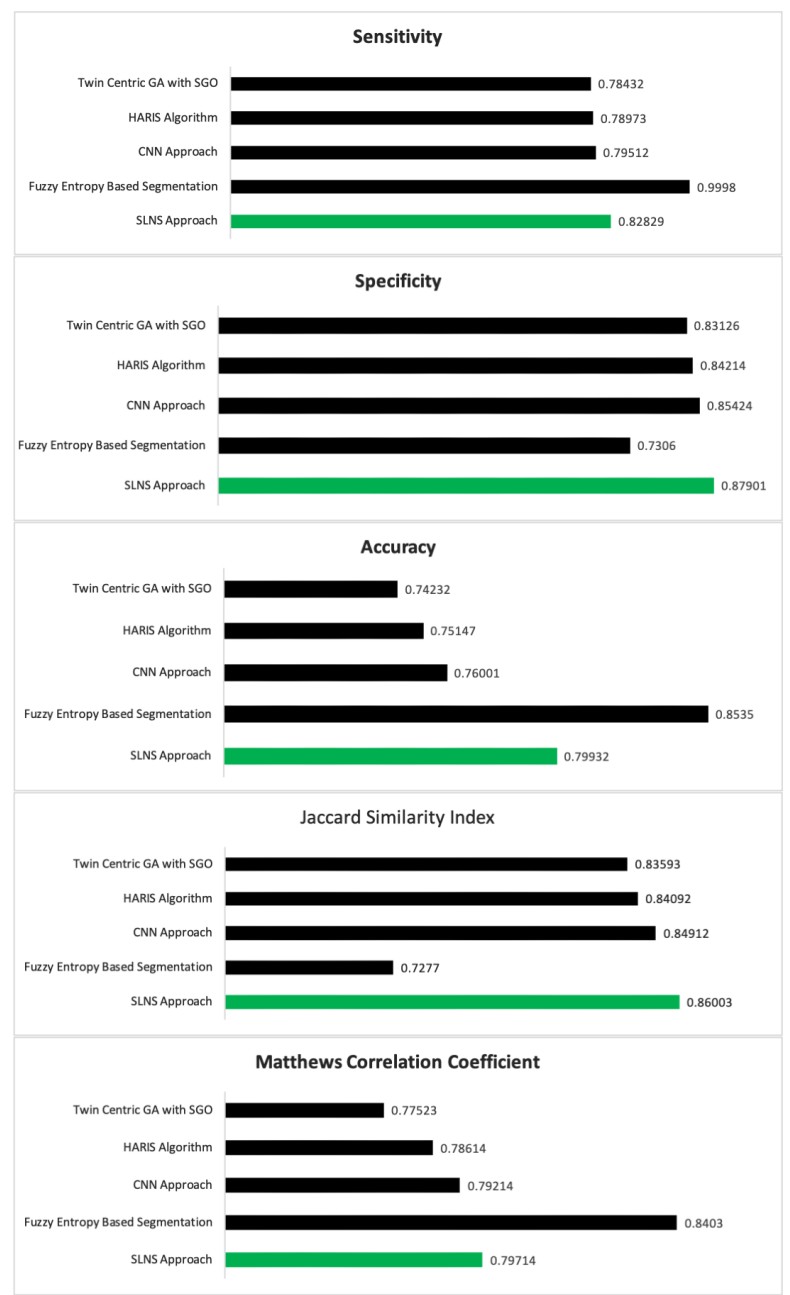

**Figure 6  The comparative analysis of various approaches.**

The enhanced tumor region presents the progress of the abnormality that will assist the physician in taking up the decisions for suitable treatment.

The self-learning-centric models are recently becoming part of the biomedical and healthcare domain, where the models are expected to work with minimal training. In a few situations where there is no adequate data available for training the model, the self-learning models are proven to perform well. The model also needs lesser training than

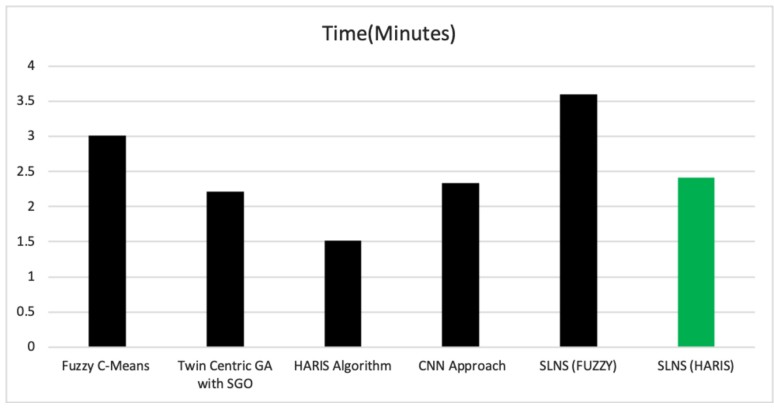

**Figure 7** **The computational time of various approaches.**

**Table 4** **The performance analysis of SLNS approach.**

|  | Tumor core (TC) | Whole tumor (WT) | Enhanced tumor (ET) |
| --- | --- | --- | --- |
| Twin centric GA with SGO | $4.778 \pm 0.781$ | $8.751 \pm 0.177$ | $7.853 \pm 0.262$ |
| HARIS algorithm | $4.825 \pm 0.098$ | $8.753 \pm 0.105$ | $7.872 \pm 0.877$ |
| CNN approach | $4.823 \pm 0.245$ | $8.756 \pm 0.714$ | $7.870 \pm 0.011$ |
| SLNS with HARIS | $4.824 \pm 0.781$ | $8.754 \pm 0.911$ | $7.874 \pm 0.982$ |
| SLNS with Fuzzy membership | $4.837 \pm 0.568$ | $8.761 \pm 0.874$ | $7.902 \pm 0.975$ |

its counterparts. The proposed model is capable of learning from its previous experimental results. The hyperparameters like the learning rate that are presented in the current study are at an acceptable level after 43 epochs, and other parameters like the training and validation studies state that the model fine-tuned the performance assessment metrics like the accuracy, sensitivity, specificity, the Jaccard Similarity Index (JSI), and the Matthew correlation coefficient (MCC) are assessed by repeated autonomous executions. The obtained values have proven that the model performs reasonably with minimal training of the data. The fuzzy component is added to evaluate the membership for assigning the pixels to the appropriate segment in contrast to the HARIS algorithm for the assignment of the pixels. The statistical analysis of the study has evinced the performance of the HARIS algorithm in the segmentation process.

## CONCLUSIONS

The pivotal object of the current study on mechanizing a self-learning model that is efficient in identifying the tumor from the MR image with minimal training. The model is efficient in working with problems that have minimal training data available. The proposed model efficiently learns from its prior experimental outcomes and utilizes the acquired knowledge for future predictions. It can be observed from the practical implementation of the proposed model that the resulting outcome seems to be very accurate and precise in identifying abnormality in the human brain. In contrast to fully supervised models like

convolutional neural networks, deep learning models and various classification algorithms need to be rigorously trained for better accuracy. Still, the proposed model is proven to be efficient in generating equivalently better outcomes with other models. The proposed model itself can normalize the noise in the input image and robust in differentiating the non-brain tissues like the skull from the brain tissues. However, the proposed approach could be further optimized by incorporating the self-correcting convolution layer through an Ancillary kernel.

The self-learning models are robust in handling the unusual problem where there is inadequate training data available. But the self-learning models overfit in few cases as the developer may not have control over the level of training to the model. The overfitting may lead to instability in the predictions that are made in few contexts. Moreover, the process of debugging the issues and rectifying them in the self-learning model is challenging. The models learn from their previous experimental outcomes, and there is a possibility that the models might misinterpret the outcome based on previous experimental results.

## Future scope

The proposed model based on the self-learning mechanism is suitable for handling uncertain data more effectively through its previous experiences. The model's performance can be further improvised by incorporating the long short term memory (LSTM) component for efficiently handling the training data for better accurate prediction of the progress in the tumor growth as the LSTM components are efficient in holding the memories for a longer time by preserving the dependencies based on the network's information. The incorporated memory elements can retain the state information over the specific iterations constructed through multiple gates. The proposed model can also be enhanced by incorporating the feedback component that would help assess tumor growth progress by correlating its previous outcome.

### Funding
The authors received no funding for this work.

### Competing Interests
Valentina E. Balas is an Academic Editor for PeerJ.

### Author Contributions
- Parvathaneni Naga Srinivasu conceived and designed the experiments, performed the experiments, analyzed the data, performed the computation work, prepared figures and/or tables, authored or reviewed drafts of the paper, and approved the final draft.
- Valentina Emilia Balas conceived and designed the experiments, performed the experiments, analyzed the data, prepared figures and/or tables, authored or reviewed drafts of the paper, and approved the final draft.

## Data Availability

Code is available in the Supplemental File.

The TCGA-LGG dataset is available at TCGA: https://wiki.cancerimagingarchive.net/display/Public/TCGA-LGG.

The dataset related to the experimentation is available at Kaggle: https://www.kaggle.com/mateuszbuda/lgg-mri-segmentation/version/1.

## Supplemental Information

Supplemental information for this article can be found online at http://dx.doi.org/10.7717/peerj-cs.654#supplemental-information.

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

deep convolutional neural network and SVM algorithm. *Computational Intelligence
Methods for Brain-Machine Interfacing or Brain-Computer Interfacing* **2020**:6789306
DOI 10.1155/2020/6789306.

**Xiao T, Xia T, Yang Y, Huang C, Wang X. 2015.** Learning from massive noisy labeled
data for image classification. In: *IEEE conference on computer vision and pattern
recognition (CVPR)*. Boston, MA, 2691–2699 DOI 10.1109/CVPR.2015.7298885.