# Peer review of "Self-Learning Network-based segmentation for real-time brain M.R. images through HARIS"

_PeerJ Computer Science, doi:10.7717/peerj-cs.654_

## Round 0.1 · original submission · Major Revisions

Check the manuscript thoroughly and attend to the following:

- Include a comparison table.
- Check and complete the graph legends for all the figures.
- Submit executable code as a supplementary file
- The cost functional used to optimize the performance of the network should be included in the manuscript.
- Read the reviewers' comments and revise the paper accordingly.

All the comments should be included in the revision.

·

Basic reporting

1
The authors introduce a lot of existing studies. To convince the future readers, the paper needs a comparison table, which uses columns for the existing studies and the manuscript, and rows for the attributes.

2
Figures 4 and 5 need complete graph legends.
Figures 4 and 5 are not cited. Please explain these figures.

3
a majority of those approaches lack inaccuracy
-> a majority of those approaches lack accuracy

4
"is presented in figure 4 below" in Line 360 should be "is presented in figure 6 below".

5
Please add the time unit (e.g., seconds or milliseconds) to Figure 6.

Experimental design

1
The authors claim that "The article's main objective
is ... computationally and technically efficient than many
other approaches like CNN and HARIS algorithm." in Abstract.
However, the computational cost of the proposed algorithm is higher than those of CNN or HARIS, according to Figure 6.
The objective of the manuscript should be clarified precisely.

2
Please explain the HARIS algorithm in detail. Moreover, please add the reference of the paper which proposes the HARIS.

Validity of the findings

1.
Your code does not work.

For example,
"from tensorflow.keras.preprocessing.image import ImageDataGnerator" in Line 9 of pcnn-Copy1.py should be
"from tensorflow.keras.preprocessing.image import ImageDataGenerator".

Please upload the correct source files and also add a readme file, which contains information about the sources and the execution procedure.

2.
In the evaluations, the authors compared the proposed algorithm with Twin Centric GA with SGO, HARIS, and CNN.
Please describe why the authors selected these algorithms to be compared.
Further, please describe why the authors did not compare many other algorithms introduced in Introduction Section.
Moreover, please describe the details of Twin Centric GA with SGO.

Reviewer 2 ·

Basic reporting

The reporting aspect of the work is up to the mark.

Experimental design

The experimental aspect is well highlighted.

Validity of the findings

The findings have been compared well with previously reported approaches.

Additional comments

The following aspects need to be addressed:
1. A few training time learning error convergence plots are required.
2. Some details regarding the approach adopted for fixing the network architecture should be incorporated.
3. At what stage of the network, does the learning saturate?
4. A confusion matrix should be used to report the best and worst time performance of the network.
5. What shall be the difference in performance with and without the fuzzy portion adopted in the work?
6. Some results relating the fuzzy entropy-based thresholding technique to the overall performance of the network should be included.
7. What cost functional is used to optimize the performance of the network?

---

## Round 0.2 · Major Revisions

The authors didn't revise the manuscript based on the previous reviewers' comments.

- Revise the manuscript based on the reviewers' comments
- All the comments have to be considered while revising the manuscript
- Upload an executable source code
- Remove the plagiarized contents
- The experiment using data augmentation suffers from data leakage
- The authors should clarify their technical contributions
- Use the latest datasets for training the proposed methodology.
- Add the missing citations

·

Basic reporting

1.
There are several description errors.
For example;
- There is a missing citation in Line 70.
- Section numbers are missing in Lines 85-87, and 146.
- "\sum_{i=0}^n" in Equations 1, 3, and 6 should be "\sum_{i=1}^n" or "\sum_{i=1}^{n-1}" because the length is n.

2.
It is impossible to calculate Equation 5. This is because we need the value of V_i to calculate V_i.

3.
The description of Page 8 is almost identical to the description of https://classic.d2l.ai/chapter_recurrent-neural-networks/bi-rnn.html
Be careful of plagiarism. Moreover, it is not necessary the explanation, because it is the basics of RNNs.

4.
Please cite and discuss several important research papers on deep neural networks using codon encoding and discuss them.
For example;
Zhang, Kaiming, et al. "CRIP: predicting circRNA–RBP-binding sites using a codon-based encoding and hybrid deep neural networks." Rna 25.12 (2019): 1604-1615.

Experimental design

1.
The experiment using data augmentation suffers from data leakage. According to the explanation of the manuscript, the dataset size was increased using data augmentation, and then the authors split the increased data into the training data and the validation data. The performance of the proposed machine learning model was evaluated using the validation data.
Therefore, the validation data and the training data can have several samples created by the same sample.

2.
Please explain why the authors select Sum, Max, Norm, EWA, and PA as the features for the input of the machine learning model.

3.
The materials submitted to PeerJ contain "cs-58149-test.json," and Table 1 in the manuscript shows the sample size of the test dataset. However, according to the manuscript, it seems that the authors did not use the test dataset.
If so, please explain why the authors did not use the test dataset.


4.
The source code "cs-58149-advanced-gru-lstm.ipynb" contains error logs.
* * *
KeyError Traceback (most recent call last)
/opt/conda/lib/python3.7/site-packages/pandas/core/indexes/base.py in get_loc(self, key, method, tolerance)
2888 try:
-> 2889 return self._engine.get_loc(casted_key)
2890 except KeyError as err:
.....
.....
.....
* * *
Please upload the correct source files.

Validity of the findings

1.
The authors constructed machine learning models that predict mRNA sequences responsible for the degradation of the COVID-19 mRNA vaccine. The structure of the machine learning models is simple and straightforward. Moreover, codon encoding is not novel in the research field on deep neural networks for RNA. The feature selection might be novel and practical, but the reason why the authors selected the features is not described well.

Therefore, the authors should clarify their technical contributions. Moreover, please describe why the authors selected the features.

Reviewer 2 ·

Basic reporting

Its is OK.

Experimental design

It has been adequately designed.

Validity of the findings

After review and modifications, the finding now stand validated.

Additional comments

The manuscript is now ready for publication.

---

## Round 0.3 · Major Revisions

Careful proofreading and spell check are required in this manuscript.
The conclusion statement should be linked to the initial research question.
Include high-resolution images
Address all the comments of the reviewer.
Revise the manuscript based on the reviewers' comments.

·

Basic reporting

The authors have satisfactorily responded to all my questions and made the necessary changes to the manuscript.

Experimental design

The authors have satisfactorily responded to all my questions and made the necessary changes to the manuscript.

Validity of the findings

The authors have satisfactorily responded to all my questions and made the necessary changes to the manuscript.

Additional comments

The authors have satisfactorily responded to all my questions and made the necessary changes to the manuscript.

Reviewer 3 ·

Basic reporting

The paper submitted investigated the use of self-learning network-based segmentation for brain MRI through HARIS and the reported methodology achieved an accuracy of 77% towards segmenting real-time images into multiple regions. The analysis workflow is interesting in general. However, please consider the following comments.

Comment 1:
A major improvement and a carefully proofread spell check are required since the English in the present manuscript is not of publication quality. Indicative:

1. Lines 21-23
2. Lines 53-58
3. Line 83
4. Lines 89-91
5. Lines 94-97
6. Lines 104-106
7. Line 133, the word “discrete” is used as a verb.
8. Lines 156-157
9. Lines 192-194
10. Lines 203-208
11. Lines 233-241
12. Lines 326-328
13. Lines 332-333
14. Lines 377-378

Comment 2:
Lines 94-97: Could you please elaborate on the following: “the issue of the massive number of segmented regions”.

Comment 3:
Lines 110-111: It is hard to understand what the authors want to state here.

Comment 4:
Lines 148-149: Please rephrase. What does computationally compatible mean? Is it computationally feasible?

Comment 5:
Line 205: Did you quantify an accepted level of noise for your data?

Comment 6:
Lines 213-216: Please rephrase the following: “for additional images upon segmenting the image.”.

Comment 7:
Lines 218-220: Please elaborate on the rationale.

Comment 8:
Lines 235-237: “… each of the convolutional layers is being guided through various techniques …”. This is quite confusing.

Comment 9:
Some figures seem to be of low resolution. Please provide images of higher quality.

Comment 10:
Figure 3 presents the loss across the epochs (as the y-axis label indicates) and not the learning rate. The authors should check if that is intended or it is misplaced.

Comment 11:
Table 1: Please include the percentage symbol somewhere in the table. I assume that numbers are percentages.

Experimental design

Comment 1:
Line 366: Did the authors use a changing/decaying learning rate? If so please elaborate and specify the strategy used. If not, the learning rate should be constant.

Comment 2:
Lines 406-409: “executing the code repeatedly for 35 images”. what do you mean? Also, what specifically is scaled up for evaluating the confusion matrix? What does “scaled up” mean?

Comment 3:
Please consider describing more carefully the research question, the rationale as well as the analysis pipeline.

Validity of the findings

The impact of the method is not clearly stated. Also, the conclusion statement should be linked to the initial research question.

---

## Round 0.4 · Minor Revisions

Read all the comments specified by the reviewer and revise the manuscript accordingly.

Correct the typo and grammatical errors

Reviewer 3 ·

Basic reporting

No comment

Experimental design

No comment

Validity of the findings

No comment

Additional comments

Thank you for taking into consideration my comments. The manuscript presents a novel segmentation architecture, however, please check again grammar and spelling mistakes.

* "to address noises in the original MRI images" -> "to address noise..."

* "renewable avenue" -> What do you mean?

* "trains and validating with ground facts" -> What do you mean?

* “layer is driven by”: Do the authors mean consists of or comprise?

* "which are slides across the original input image" -> slides of the image?

* "Over numerous rounds, the technique is repeated over multiple iterations to identify all the key features in the input image" -> What do the authors mean? Does this refer to the training epochs?

* "The adaptive structural similarity index(ASSI) approach" -> Is this a metric and not an approach?

* "Overfitting occurs ...., resulting in improper categorization of the abnormal area due to an abundance of data". -> Why this is happening? Please elaborate on that, otherwise please rephrase.

* "The hyperparameters like the learning rate that are presented in the current study are at an acceptable level after 43 epochs, and other parameters like the training and validation studies state that the model fine-tuned” -> What do the authors mean by learning rate is at an acceptable level?

From the technical perspective: what happens if unlabeled data further used as new training samples are misclassified? Also, what is the minimum amount of acceptable labelled data to initially train the model?

minor: I do suggest the use of "MR images" instead of MRI images since "I" corresponds to imaging.

---

## Round 0.5 · accepted · Accept

Congratulations...I am informing you that your manuscript - Self-Learning Network-based segmentation for real-time brain M.R. images through HARIS - has been Accepted for publication.

Reviewer 3 ·

Basic reporting

no comment

Experimental design

no comment

Validity of the findings

no comment

Additional comments

All comments have been properly addressed. I would appreciate if authors will elaborate a little bit on the following (included in Peerj_Rebuttal.docx) since all of my comments highlighted specific parts of the document with text that was copied from the original manuscript.

"...The recommendation of the reviewer seems to be misplaced with other manuscript of peerj publications. Hence majority of the comments seems to be irrelevant to the current manuscript...."